# Di- and Monorhamnolipids Produced by the *Pseudomonas putida* PP021 Isolate Significantly Enhance the Degree of Recovery of Heavy Oil from the Romashkino Oil Field (Tatarstan, Russia)

**Liliya Biktasheva, Alexander Gordeev, Svetlana Selivanovskaya and Polina Galitskaya \***

Institute of Environmental Sciences, Kazan Federal University, 18 Kremlyovskaya St., 420008 Kazan, Russia; biktasheval@mail.ru (L.B.); drgor@mail.ru (A.G.); svetlana.selivanovskaya@kpfu.ru (S.S.)

\* Correspondence: gpolina33@yandex.ru; Tel.: +7-903-340-40-23

**Abstract:** Around the globe, only 30–50% of the amount of oil estimated to be in reservoirs ("original oil in place") can be obtained using primary and secondary oil recovery methods. Enhanced oil recovery methods are required in the oil processing industry, and the use of microbially produced amphiphilic molecules (biosurfactants) is considered a promising efficient and environmentally friendly method. In the present study, biosurfactants produced by the *Pseudomonas putida* PP021 isolate were extracted and characterized, and their potential to enhance oil recovery was demonstrated. It was found that the cell-free biosurfactant-containing supernatant decreased the air–water interface tension from 74 to 28 mN m$^{-1}$. Using TLC and FTIR methods, the biosurfactants produced by the isolate were classified as mono- and di-rhamnolipid mixtures. In the isolates' genome, the genes *rhlB* and *rhlC*, encoding enzymes involved in the synthesis of mono- and di-rhamnolipids, respectively, were revealed. Both genes were expressed when the strain was cultivated on glycerol nitrate medium. As follows from the sand-packed column and core flooding simulations, biosurfactants produced by *P. putida* PP021 significantly enhance the degree of recovery, resulting in additional 27% and 21%, respectively.

**Keywords:** enhanced oil recovery; biosurfactant; mono- and di-rhamnolipids; sand-packed column simulation of oil recovery; core flooding simulation of oil recovery

## 1. Introduction

Oil recovery, which is the extraction of crude oil from an oil field, can be subdivided into primary, secondary, and tertiary, or enhanced. Primary oil recovery occurs due to natural pressure in an oil reservoir using simple mechanical pumps, while secondary recovery techniques include artificial creation of additional pressure in the oil reservoir using injection water. Around the globe, only 30–50% of the amount of oil estimated to be in reservoirs ("original oil in place") can be obtained using primary and secondary oil recovery methods [1]. Enhanced oil recovery methods usually include injection of steam (thermal methods), gas, or chemicals to decrease oil viscosity. These methods enable the extraction of 10–40% more oil over the initial amount [2]. Recently, alternative methods with higher efficacy, lower cost, and lower environmental impact for enhanced oil recovery have been required.

The Romashkino oil field situated in Tatarstan (Russia) has a large oil field area, a small formation dip, many reservoir layers, a wide oil-water transition zone, and a complicated sedimentary environment. It has been exploited since 1948, and the oil recovery degree since then has decreased from 53% to 45%. Because of its high oil viscosity (2.6–

4.5 mPa·s) and high sulfur and wax contents (1.3 and 3.2%, respectively), the Romashkino oil field urgently needs the development and implementation of new efficient methods for enhancing oil recovery [3,4].

Biosurfactants are amphiphilic microbial products that may be used for enhanced oil recovery since they can reduce crude oil–water interfacial tension (IFT) and alter emulsification and wettability. Biosurfactants possess lower toxicity and higher biodegradability than chemical surfactants. Due to their relatively lower efficacy (in comparable concentrations) and higher costs of production, biosurfactants are still not widely used in large-scale oil production or in the remediation of polluted sites [5–7]. However, new biosurfactant producers and new biosurfactant types with improved characteristics are often being discovered, providing good prospects for the future commercial use of biosurfactants for enhanced oil recovery, considering the growing costs of chemical biosurfactants and the growing awareness of their negative impact [5,8–12]. It should be noted that there is a lack of publications concerning specific aspects of biosurfactant usability for enhanced oil recovery, e.g., information available about the recovery degree of heavy oils in complicated sedimentary environments such as in the Romashkino oil field situated in Tatarstan, Russia [13].

The hydrophobic component of biosurfactants usually consists of long-chain fatty acids or saturated or unsaturated hydrocarbons, while the hydrophilic component is made up of organic acids, alcohols, or other carbohydrates [14,15]. Thus, according to their chemical structure, biosurfactants are classified into glycolipids, lipopeptides, phospholipids, neutral lipids, substituted fatty acids, and polysaccharides [16,17].

Biosurfactant producers, which include yeast, fungi, and mainly bacteria, originate from oil-polluted aquatic and soil environments as well as from the rhizosphere or other plant-associated locations [16,18,19]. Biosurfactants are produced by microbes for different goals, e.g., for transportation across membranes, for protection from predators, parasites and competitors, for interaction with hosts, for the formation of biofilms, and for changes in the properties of environments and nutrient sources, such as wetting and metal sequestration [12,20,21]. In environments containing hydrocarbons, microbes produce biosurfactants to provide interactions between cell physiology, the cell surface, and hydrocarbons that are substrates for the cell. Thus, low molecular mass biosurfactants are used to solubilize poorly soluble hydrocarbons by means of micelles and aggregate formation, while high molecular mass biosurfactants emulsify the hydrocarbons in water medium, and they may be bound to the cell wall, modifying the membrane, and enabling hydrocarbons to pass across the wall. It has been reported that some types of biosurfactants stimulate the growth of their producing strains, playing a vital role in the interaction between microbes and their environments. However, many mechanisms of biosurfactant synthesis and many goals that microbes achieve using biosurfactants have not yet been determined [8].

Rhamnolipids are well-studied biosurfactants due to their high emulsifying activity as well as high yields [22–25]. In large-scale bioreactors, from 1 to 10–12 and in some cases to 30–50 g $L^{-1}$ rhamnolipids may be obtained utilizing different types of oils as a carbon source [26–29]. Structurally, rhamnolipids consist of one or two rhamnoses and one or two β-hydroxyl fatty acids of different carbon chain lengths ($C_8$-$C_{10}$, $C_{10}$-$C_{10}$, $C_{10}$-$C_{12}$, $C_{10}$-$C_{12:1}$ etc.). It has been reported that rhamnolipids with different structures possess different activities; thus, mono-rhamnolipids have been demonstrated to be more inhibitory towards plant fungal pathogens, and their emulsifying activity is higher than that of di-rhamnolipids [9,30–32].

Rhamnolipids are produced by some representatives of *Pseudomonas*, *Dietzia,* and other bacterial genera, while the pathogenic *P. aeuroginosa* is the most well studied efficient producer [32–34]. Non-pathogenic *Pseudomonas* species such as *P. putida*, *P. citronellolis*, *P. cepacian,* and other *Pseudomonas* spp. are less mentioned as potential rhamnolipid producers in the scientific literature [35–39]. Mono-rhamnolipid synthesis in bacteria involves the rhamnotransferase RhlB, while dTDP-L-rhamnose and β-hydroxyl fatty acids are used as precursors. Di-rhamnolipids are synthesized using dTDP-L-rhamnose, and

mono-rhamnolipids are precursors involving rhlC rhamnotransferase. In addition, rhamnolipid production is controlled by genetic regulation of the *rmlBDAC* and *rhlAB* operons. It was demonstrated that the transformation of *Pseudomonas* strains with plasmids carrying *RhlB* and *RhlC* encoding genes enabled them to produce mono- and di-rhamnolipids respectively [32]. Interestingly, in most publications cited above, rhamnolipids of *Pseudomonas* origin are comprehensively characterized in terms of their chemical composition and ability to emulsify oil, collapse drops, inhibit pathogens, and alter the interface tension but not in terms of their ex situ capacity to enhance oil recovery, in particular heavy oil.

However, taking into account the promising abilities of rhamnolipids as surfactants produced by *Pseudomonas*, they are good candidates to be used in microbially enhanced oil recovery. The objective of the present study was to obtain and characterize biosurfactants produced by *P. putida* PP021 isolated from oil-polluted soil and to investigate their potential to enhance the recovery of heavy crude oil from the Romashkino oil field using two ex situ techniques: sand-packed column and core flooding.

## 2. Materials and Methods

Biosurfactant producing strain *P. putida* PP021 was isolated from an old oil spill situated in the production area of the Romashkino oil field (Tatarstan, Russia). The strain was identified on the basis of 16S rRNA gene sequencing and stored in the laboratory museum of the Institute of Environmental Sciences of Kazan Federal University (Russia).

To estimate the biosurfactant production activity, the strain was cultivated in glycerol nitrate medium at 35 °C and 180 rpm for 1–6 days. Each day, the optical density of the growing culture was measured at 600 nm, and the cell-free supernatant was obtained by centrifugation at 8000 rpm for 10 min. Interfacial tension was measured using a KS20 (Kruess, Hamburg, Germany) tensiometer using the Dui Nui method.

In order to obtain the biosurfactants, the cell-free supernatant obtained as described above was pH-adjusted using 2 N HCl, incubated overnight at 4 °C and centrifuged at 10,000 rpm and 4 °C for 20 min. The precipitated fraction was dissolved in a chloroform:methanol (2:1, *v/v*) mixture, and the crude biosurfactant mixture was obtained using a rotary evaporator under vacuum. The yield of the acid precipitated fraction of the biosurfactants (APF) was estimated to be 9.8 g from 1 L of cell-free supernatant.

Characterization of biosurfactants was conducted using two methods—thin-layer chromatography (TLC) and Fourier transform infrared spectroscopy (FTIR).

Separation of different rhamnolipid fractions was performed by employing TLC. To assess the composition of the biosurfactants obtained, the crude mixture was dissolved in CHCl3/CH3OH (1:1 *v/v*) and spotted on a silica gel plate (G60, Merck, Darmstadt, Germany). The TLC solvent was a mixture of chloroform:methanol:ammonia solution (65:35:5, *v/v/v*). The Rf values of the spots obtained were calculated from the TLC plate exposed to UV light. To detect sugars, lipids, and free amino groups in the TLC spots, the following chromogenic agents were used: (i) 100 mL of acetic acid supplemented with 2 mL of sulfuric agent and 1 mL of p-anisaldehyde, (ii) iodine vapor, and (iii) 1% ninhydrin reagent. After staining, the plates were incubated at 110 °C for 10 min for spot development. Mono- and di-rhamnolipid standards (Sigma Aldrich, Burlington, MA, USA, 99% pure) were used to identify different fractions of rhamnolipids on the TLC plates [40].

To determine the type and structure of biosurfactants, Fourier transform infrared spectroscopy was performed using LUMOS I (BRUKER, Billerica, Massachusetts, USA) for the APF. The spectra were collected from 400 to 4000 wavenumbers (cm$^{-1}$).

The ability of the bacterial strain to synthesize mono- and di-rhamnolipids was analyzed using real-time PCR with specific primers for the *rhlB* and *rhlC* genes, respectively (forward and reverse, 5′-3′: GCCCACGACCAGTTCGAC and CATCCCCCTCCCTATGAC, CCGAAGCTTATGAGCGGCCTGTTCCACT and CTTGGAATTCCCGGAAGCTACGGACG). DNA was extracted from the pellet of the overnight culture after centrifugation using the FastDNA™ SPIN Kit for Soil (MP Biomedicals) according to the manufacturer's instructions. Real-time PCR was carried out with SYBR



Green in a total volume of 20 μL that contained 1X PCR buffer, 0.2 mM dNTPs, 1 μM of each primer, 2 mM MgCl$_2$, 1.25 U Taq DNA polymerase, and 1 μM DNA. Amplification was carried out in a CFX 96 (Bio-Rad Laboratories, Inc., Hercules, California, USA) thermocycler using the following program: denaturation at 94 °C for 2 min, 35 cycles of 94 °C for 30 s, 58 °C for 1 min, extension at 72 °C for 1.5 min, and final extension at 72 °C for 7 min [9,41,42]. To analyze the expression levels of the *rhlB* and *rhlC* genes, RNA was extracted from the isolate and converted to cDNA.

Before RNA extraction, the strain was cultivated (i) in glycerol nitrate medium at 35 °C and 180 rpm for 6 days and (ii) in LB medium under the same conditions. RNA was extracted using the RNeasy PowerSoil Total RNA Kit and RNase-Free DNase Set (Qiagen, Hilden, Germany). RNA was converted to cDNA using the Transcriptor High Fidelity cDNA Synthesis Kit (Roche, Basel, Switzerland). cDNA was further used in the real-time PCR procedure described above. The efficiency of each primer set was calculated in Bio–Rad CFX 96 software using a calibration curve obtained from 10-fold serial dilutions of cDNA. Relative expression levels were calculated with the Ct values obtained and corresponding primer efficiencies as described by Pfaffl and co-authors [43].

The potential of APF to enhance tertiary oil recovery was investigated in two types of experiments: sand column experiments and core flooding experiments. Crude oil for both experiments was obtained from the oil well number 2313 of the Romashkino oil field (Tatarstan, Russia) in June 2021. The crude oil had the following characteristics: density—0.91 g cm$^{-1}$, viscosity—3.7 mPa·s, wax content—3.2%. In both experiments, a solution of APF in brine (NaCl, 5 wt%) with a concentration of 100 mg L$^{-1}$ was used.

For the column experiment, twenty grams of soil mixture (sieved agricultural soil/sieved sand 1:1 *w:w*) was packed into a glass column with a diameter of 1.5 cm and a length of 25 cm. The soil contained 32%, 26%, and 42% of clay, silt, and sand, respectively, the content of organic matter was 3.1 g kg$^{-1}$ and the pH was 7.1. The clean river sand was air-dried and sieved with a 2 mm mesh. Eight centimeters of the column on the top remained empty to enable the addition of liquids. The column was filled stepwise with brine until the volumes of added and released water had equalized. The difference between the total volumes of added and exceeded water was assumed to be the pore volume (PV). Furthermore, crude oil was forced to pass through the column, and the volume of brine released from it was assumed to be the initial oil in place. The column was incubated for 1 day at room temperature, and 6 PV of brine was added to the column in a stepwise manner to simulate secondary oil recovery. The total amount of oil released from the column after brine flooding was registered. Furthermore, 6 PV of APF solution in brine was added to the top of the column, and the column was incubated for 3 days at room temperature. An additional 10 PV of brine was added stepwise to the column to simulate enhanced oil recovery. The total volume of oil released from the column after APF solution and the second brine addition was registered. The rates of secondary and tertiary oil recovery relating to the initial oil in place amount were calculated as percentages [44].

For the core flooding experiment, the core was obtained from the producing reservoir of the Romashkino oil field located in Tatarstan (Russia). It was cleaned in a Soxhlet apparatus (BUCHI, Flawil, Switzerland) using a chloroform:methanol mixture (75:25) and dried at 80 °C for 24 h. The cleaned core was weighed and fixed in the core holder of the flooding system presented in Figure 1. The fixed core was vacuumed for 2 h to remove pore water and air. Then, it was filled with brine at a flow rate of 0.5 mL min$^{-1}$ for 12 h, and the weight of the saturated core was determined. The difference between the two core weights before and after saturation reflected the PV of the core (PV). Subsequently, crude oil was injected into the core at a flow rate of 0.3 mL min$^{-1}$ for 24 h, and the volume of brine displaced by the crude oil was registered to calculate the original oil in place amount. Afterward, the core was incubated for 3 days. To simulate the process of water flooding, brine was injected at a rate of 0.5 mL min$^{-1}$ until no more oil flowed out from the core, and the volume of oil was determined in the flooding process to estimate the sec-

ondary oil recovery. To simulate tertiary oil recovery, APF solution in brine (approximately 6 PV) was added to the core at a rate of 0.3 mL min$^{-1}$ until no more oil flowed out, and the volume of oil was registered in the flooding process [45,46]. All manipulations with the core were conducted at 37.8 °C, as this temperature was reported to be typical for the Romashkino oil field [3].

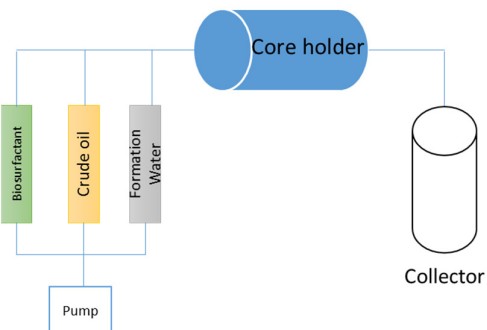

**Figure 1.** Scheme of core flooding equipment.

Air–water tension assessment, and the determination of gene expression was conducted in four independent cultures analyzed in triplicate. Corresponding figures contain the means and standard deviations. The qPCR results were analyzed using Student's *t* test, and other results were analyzed using the Mann–Whitney U test. All statistical analyses were performed at a significance level of 0.05.

## 3. Results

### 3.1. Air–Water Interface Tension

Supernatant obtained in the process of *P. putida* PP021 incubation led to alteration of the air–water surface tension (Figure 2a). Immediately after bacterial inoculation, the surface tension of the supernatant was estimated to be 73.8 mN m$^{-1}$, while on the 3rd and following days after inoculation, it decreased to ~27 mN m$^{-1}$. The results obtained are in line with those presented in the scientific literature for other efficient biosurfactant producers of *Pseudomonas* origin and suggest the high potential of *P. putida* PP021 to produce biosurfactants for enhancing oil recovery [36,47]. It should be noted that the optical density of bacteria reached a plateau only on the 5th day of incubation (Figure 2b). At least three reasons might explain this absence of correlation between the bacterial density and the activity of biosurfactants observed. First, bacteria could stop producing biosurfactants; second, the ratio between different types of biosurfactants (that possess different properties) could change; and third, surfactant molecule saturation could be reached at the interface [9,40]. Regardless, the relative alteration of surface tension per unit of bacterial density, or per cost of incubation duration, is an important economic characteristic for the commercial production of biosurfactants.

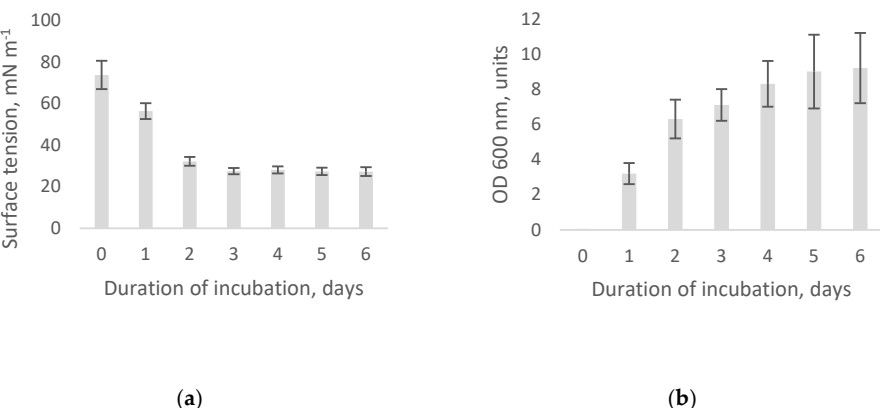

(**a**)                                   (**b**)

**Figure 2.** Surface tension of (**a**) and optical density (**b**) of the *P. putida* PP021 cultural medium.

### 3.2. TLC and FTIR

The biosurfactants produced by *P. putida* PP021 were extracted and purified to characterize their chemical structure. After purification, TLC was conducted to separate different surfactants. Exposure of the TCL plate to UV revealed three main spots (Figure 3a) with retention factors (Rf) of 0.79, 0.74, and 0.36. Further, the TLC plate was treated with iodine vapor, ninhydrine, and anisaldehyde to reveal the basic chemical nature of the biosurfactants (Figure 3b–d). The results suggest that proteins were absent, while carbohydrates and lipids were present in two of three spots (with Rf = 0.74 and Rf = 0.36); therefore, these two spots contain biosurfactants belonging to glycolipids. Indeed, bacteria of the *Pseudomonas* genus are reported to be able to produce biosurfactants from the glycolipid class that contain one or two rhamnoses and fatty acids of different C-chain lengths [10,30]. The two spots therefore may be considered mono- and di-rhamnolipids.

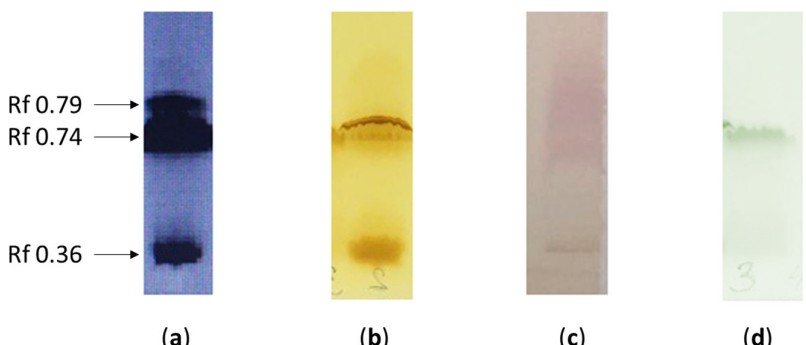

(a)                  (b)                  (c)                  (d)

**Figure 3.** Characterization of the chemical structure of the biosurfactant using TLC: (**a**) spots under UV light, (**b**) spots after iodine staining, (**c**) spots after ninhydrin staining, (**d**) spots after anisaldehyde staining.

These results were confirmed by analysis of the FTIR spectrum of the APF (Figure 4). Hydroxyl groups identified by symmetric O–H stretching referred to carbohydrate fragments, the region after 1648 $cm^{-1}$ to 1316 $cm^{-1}$ corresponded to groups characteristic of carbonyl groups in unsaturated aliphatic carboxylic acids, a weak C=O ester signal at 1724 $cm^{-1}$ and fan stretching vibration for $CH_2$. The observed peaks correspond to the spectra described by other authors for rhamnolipids [48–50].

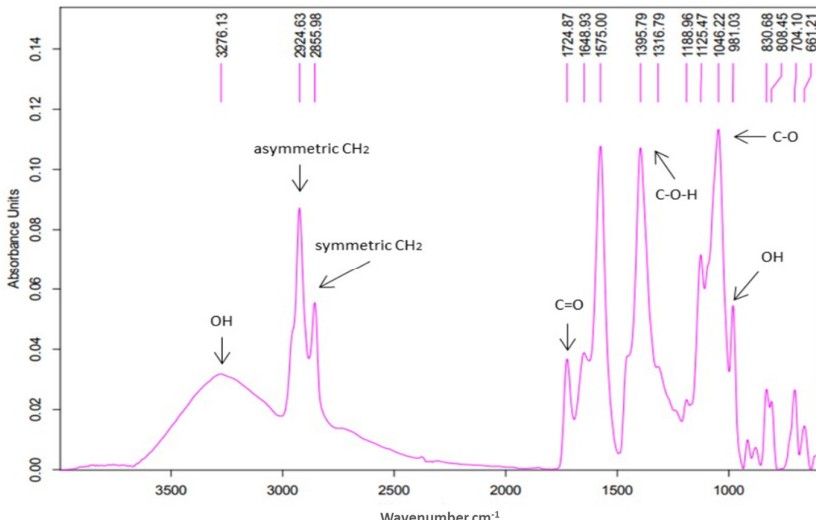

**Figure 4.** FTIR spectrum of the APF.

### 3.3. Presence and Relative Expression of Genes Encoding Rhamnotransferase Synthesis

The results of chemical characterization of the surfactants produced by *P. putida* PP021 were confirmed using molecular biology methods. The presence and expression of genes encoding enzymes involved in the rhamnolipid synthesis were analyzed using RT–PCR and reverse transcription RT–PCR techniques (Figure 5). It was revealed that both the *rhlB* and *rhlC* genes, encoding rhamnosyltransferase 1, which catalyzes dTDP-L-rhamnose, and β-hydroxy fatty dTDP-L-rhamnose accessions, which compose mono-rhamnolipis, and rhamnosyltransferase 2, which catalyzes dTDP-L-rhamnose and mono-rhamnolipid accessions, were present in the genome of the isolate (data not shown). Both genes were also expressed when the isolate was cultivated in glycerol nitrate medium for 6 days. The relative expression level of the *rhlB* gene was higher than that of *rhlC*. Indeed, rhamnosyltransferase 1 is involved in the synthesis of both mono- and di-rhamnolipids, while rhamnosyltransferase 2 is involved only in di-rhamnolipid synthesis [9]. We assume that this confirms the presence of both types of rhamnolipids in the cell-free supernatants of the isolate. Interestingly, the relative expression levels of the *rhlB* and *rhlC* genes were significantly higher ($p < 0.05$) when the isolate was grown in a medium with glycerol as a sole carbon source than when it was grown in the nutrient-rich LB medium. This confirms that rhamnotransferases 1 and 2 are inducible but not constitutive enzymes and suggests that the yield of rhamnolipids produced by the *P. putida* PP021 strain can be increased by optimization of the incubation conditions.

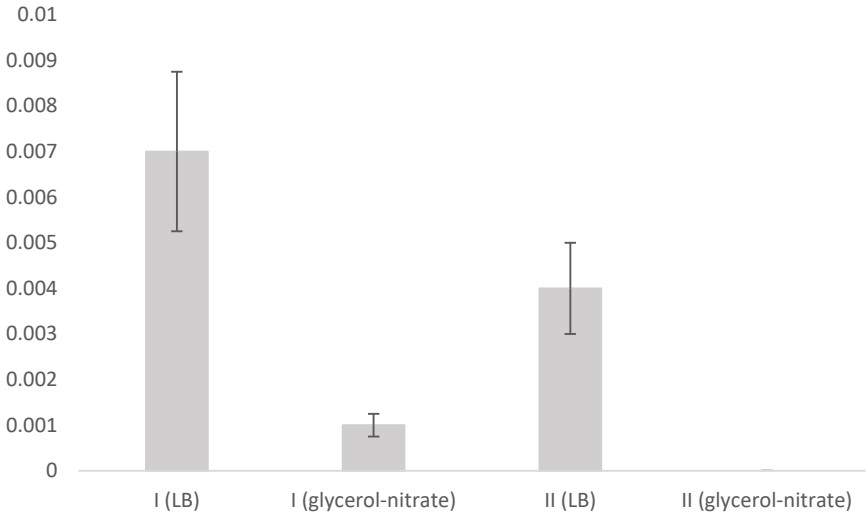

**Figure 5.** Relative expression of the *rhlB*-encoding rhamnotrasferase 1 (I) and *rhlC*-encoding rhamnotransferase 2 (II) genes in *P. putida* PP021 grown on LB and glycerol nitrate media.

### 3.4. Enhanced Oil Recovery

The efficacy of the APF to enhance oil recovery was assessed using two simulation methods: sand-packed columns incubated at room temperature and sandstone cores incubated at 37.8 °C, which corresponds to the average temperature in the oil reservoir number 2313. The sand-packed column method has the advantage of reproducibility and the possibility to conduct many experimental variants at the same time (e.g., with different biosurfactant concentrations or with different types of surfactants) and therefore to compare them correctly. The disadvantage of this method lies in the lower surface area of the column's material in comparison with the original core and the consequent exaggerated value of the enhanced oil recovery. In contrast, the use of a natural reservoir core in the reservoir temperature conditions allows practically relevant values of enhanced oil recovery to be obtained. However, it does not allow the comparison of several methods of enhancement or to conduct experiments in many replicates to obtain statistically significant results [44–46]. Thus, the combination of the two simulation methods might overcome disadvantages of both of them and provide relevant conclusions.

In the sand-packed column experiment, brine was forced to pass through the column, and using the difference of added and excessive brine, the PV was determined to be 6.21 mL. After crude oil was forced into the column with the brine-saturated pores, a volume of brine excessed from it (3.81 mL). This volume was assumed to be the amount of original oil in place. Original oil in place occupied approximately 61% of the PV. Later on, the degrees of oil recovery were calculated in relation to this amount. The secondary oil recovery simulated by flooding of the column with brine resulted in 1.03 mL of oil (27%). The enhanced oil recovery simulated by treatment with biosurfactant solution led to the elution of 1.83 mL of oil (i.e., an additional 48%).

In the core flooding experiment, the PV was estimated to be 12.67 mL, and the original oil in place amount was 6.30 mL. Thus, the original oil in place occupied approximately 50%, which is lower than that in the column experiment. Presumably, sand and soil particles of the artificial column had larger surface areas than the original core, and therefore, more oil was absorbed there on the particles and in the pores. As shown in Figure 6, secondary oil recovery increased intensively, shortly after brine injection. Furthermore, the recovery slowed down, and after the injection of 2.5 PV, the recovery degree remained stable independent of brine injection. The secondary oil recovery degree in the core flooding experiment was estimated to be 40–42% of the amount of original oil in place, which was significantly higher than that in the column experiment. Presumably,

the oil in the core was contained in the pores and was not as bound to particles as in the column experiment. The addition of the APF led to a 20–22% additional increase in oil recovery. This result can be assessed as efficient compared with reports of other authors, especially considering the type of crude oil that was used in the experiment—heavy oil—which has high wax and sulfur contents [5]. The mechanism of biosurfactant efficacy lies in its ability to alter the interface tension since it increased the capillary number, which in turn lowered the residual oil saturation [51]. The result of tertiary oil recovery enhancement obtained in this study is also efficient compared with conventional chemical flooding agents such as polymers, surfactants, composites, and alkali. Thus, chemical flooding is considered to be efficient if over 10% of the original oil in place is recovered in addition to the initial water flooding [52].

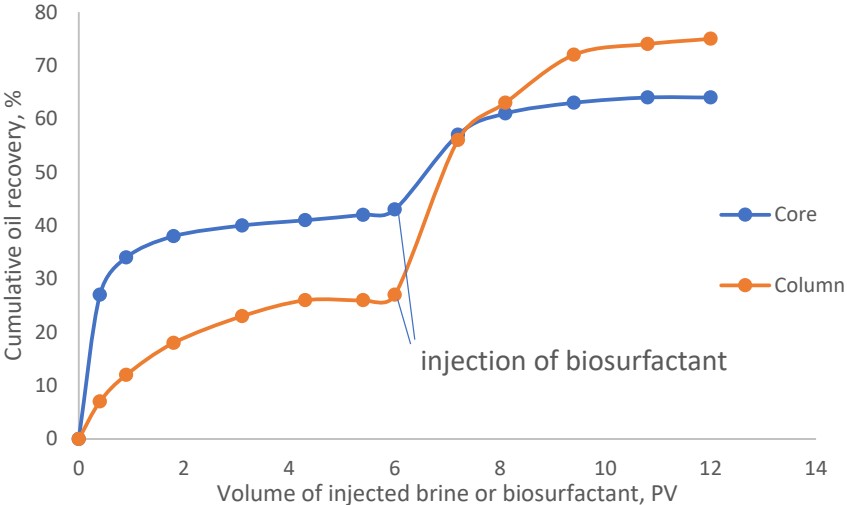

**Figure 6.** Secondary (before biosurfactant injection) and enhanced (after biosurfactant injection) oil recovery as estimated in sand-packed column and core flooding experiments.

## 4. Conclusions

It can be concluded that biosurfactants produced by the newly isolated strain *P. putida* PP021 contain compounds belonging to the di- and mono-rhamnolipid groups. The cell-free supernatant obtained from the cultural medium after 6 days of cultivation of the isolate in the glycerol nitrate medium decreased the air–water interface tension from 73.8 to 27.3 mN m$^{-1}$. The APF produced by the isolate enhanced the degree of recovery of heavy crude oil from the Romashkino oil fields (Russia), by 48% and 21% as observed in the oil in sand-packed column and core flooding experiments, respectively.

**Author Contributions:** Conceptualization, P.G. and S.S.; methodology, L.B.; validation, A.G.; formal analysis, P.G.; investigation, L.B. and A.G.; resources, L.B.; data curation, S.S.; writing—original draft preparation, S.S.; writing—review and editing, P.G.; visualization, A.G.; supervision, S.S.; funding acquisition, P.G. All authors have read and agreed to the published version of the manuscript.

**Funding:** This work was supported by the Ministry of Science and Higher Education of the Russian Federation under agreement No. 075-15-2020-931 within the framework of the development program for a world-class Research Center "Efficient development of the global liquid hydrocarbon reserves".

**Institutional Review Board Statement:** Not applicable.

**Informed Consent Statement:** Not applicable.

**Data Availability Statement:** Data is contained within the article.

**Conflicts of Interest:** The authors declare no conflict of interest. The funders had no role in the design of the study; in the collection, analyses, or interpretation of data; in the writing of the manuscript, or in the decision to publish the results.

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
