# Peer review of "Di- and Mono-Rhamnolipids Produced by the Pseudomonas putida PP021 Isolate Significantly Enhance the Degree of Recovery of Heavy Oil from the Romashkino Oil Field (Tatarstan, Russia)"

_processes, doi:10.3390/pr10040779_

Round 1
Reviewer 1 Report
The paper investigates the potential of application of biosurfactants on oil recovery. The results are interesting and the authors provide comprehensive data to support the conclusion. I recommend to publish the paper once the following concerns are addressed.
- What is the yield of biosurfactants during the culture of bacterium. Did culture condition affect the yield and performance of the surfactants?
- The characteristics of sandpack , core and crude oil need to be provided.
- The difference of purpose of conducting core flooding and sandpack need to be discussed.
Author Response
Dear Reviewer, thank you for the accurate reading and comments to our article. Please find our response in the attached file

Reviewer 2 Report
The overall research is good and presented well. Please check subscript/ superscript throughout the paper also unit of a litre is L, not l, correct it wherever it is necessary.
Author Response
Dear Reviewer, thank you for the accurate reading and comments to our article. We made corrections according to them. Please find the details in the attached file

Reviewer 3 Report
The authors investigated the potential of biosurfactants produced by an isolate to enhance oil recovery. The data of surface tension, FTIR and enzyme analyses revealed the mechanisms of biosurfactant production and how they enhanced the oil recovery. I addition, a sand column test and a core flooding simulation were conducted to demonstrate the enhancement in oil recovery. The evaluation is quite systematic, and the data are sufficient to support the conclusions. I would recommend the following to improve the quality of the manuscript: (1) the quality of all figures can be improved. For example, the functional groups can be shown on spectrum of FTIR. (2) additional polishing on the English writing.
Author Response

(The authors gave the same response as above.)
